# Investigation of Reverse Recovery Current of High-Power Thyristor in Pulsed Power Supply

**Jiufu Wei**[ID]**, Zhenxiao Li and Baoming Li \***

National Key Laboratory of Transient Physics, Nanjing University of Science and Technology, Nanjing 210094, China; wjfnjust@163.com (J.W.); lizhxnjust@126.com (Z.L.)

**\*** Correspondence: baomingli@njust.edu.cn; Tel.: +86-139-5181-9662

**Abstract:** The instantaneous overvoltages from the load side can cause damages of high-power thyristors in conventional pulsed power supply topologies, especially in cases of numerous pulse-forming units that operate together with discharge time intervals. The instantaneous overvoltages from the load side, which leads to high reverse recovery currents in high-power thyristors, can be induced by load mutations in the electromagnetic launching field. This paper establishes circuit models of PPS topologies, and investigates effects of the initial voltage of the energy-storage capacitor, the discharge time intervals, and the load resistance on the reverse recovery currents in high-power thyristors. To overcome the shortcomings of conventional PPS topologies, an improved PPS topology is developed. The improved PPS topology applies coupling inductor and resistance-capacitance snubber techniques, which can absorb the surge energy from the load side and reduce the reverse recovery currents in high-power thyristors. The simulation technique has been applied to validate theoretical analysis and the proposed model.

**Keywords:** PPS; high-power thyristors; reverse recovery currents; electromagnetic launching field

## 1. Introduction

Pulsed power supply (PPS) is widely applied in the production of nano-powder [1], drilling of hard rocks [2], electrothermal-chemical gun [3], electromagnetic railgun [4–10], etc. There are various energy storage ways for PPS, such as capacitor bank, homopolar generator-inductor, explosive magnetic flux compression [11]. The PPS investigated in this paper is based on the capacitor bank, which is mainly composed of high-power energy-storage capacitors (hereinafter referred to as energy-storage capacitors) [4], high-power semiconductor devices [12–20], and high-power pulse-shaping inductors (hereinafter referred to as pulse-shaping inductors). High-power semiconductor devices include high-power thyristors (hereinafter referred to as thyristors) [12–17] and high-power fast recovery diodes (hereinafter referred to as fast recovery diodes) [18–20], as shown in Figure 1.

The investigated PPS is widely applied in the electromagnetic launching (EML) field [3–10]. There are two types of conventional PPS topologies, which are defined as type I and II PPS topologies in this paper, respectively. Each PPS topology contains multiple pulse-forming units (PFUs) in parallel in practical applications. Compared with the devices such as energy-storage capacitors and pulse-shaping inductors, the semiconductor devices in conventional PPS topologies are easier to suffer from overvoltage and high current change rate $di/dt$ in the reverse recovery phases, which inevitably leads to damages of them [6]. The overvoltage and high current change rate $di/dt$ can be induced by load mutation in the EML experiment [5]. The load mutation can lead to an open circuit or sharp increase in resistance of the load side, which results in instantaneous overvoltage and high current change rate $di/dt$ in parts of pulse-shaping inductors. Due to the randomness of load mutation, a typical phenomenon in a multiple uninterruptible discharge process is that the thyristors in conventional PPS

topologies are not always but randomly damaged, especially in the case of numerous PFUs operating together with discharge time intervals, although all the initial conditions and devices' parameters keep the same.

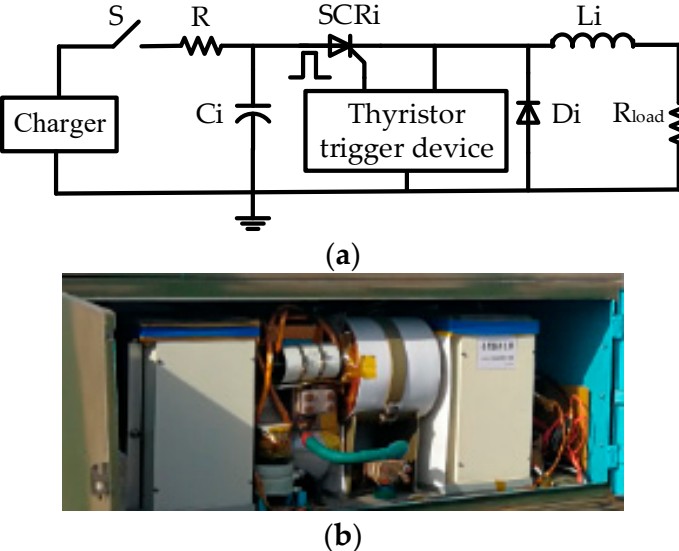

**Figure 1.** Type I pulse-forming unit (PFU) topology and its photograph. (**a**) Type I PFU topology; (**b**) photograph of type I PFU topology.

This paper establishes circuit models of PPS topologies. Then, the reverse recovery current characteristics of the thyristors are explored. Effects of the initial voltage of the energy-storage capacitor, the discharge time intervals, and the load resistance on the reverse recovery currents are systematically investigated. It is found that the reverse recovery currents in the thyristors in the type I PPS topology are larger than those in the type II PPS topology under the same working conditions, indicating that the type II PPS topology is more helpful to reduce the breakdown risks of the thyristors if multiple PFUs operate together with discharge time intervals.

However, the reverse recovery currents in the thyristors in both type I and II topologies are relatively high when sustaining instantaneous overvoltages caused by load mutations. Therefore, the reliability of conventional PPS is relatively low in the worst cases and it is imperative to enhance its robustness.

References [6,19] indicate that adopting a resistance-capacitance (RC) snubber circuit in parallel with the thyristors can protect them from being damaged. It can reduce the reverse recovery currents but consumes more useful energy because it is more sensitive to the voltage variation of the thyristors and less sensitive to the voltage variation of the load side. If multiple PFUs operate together with discharge time intervals, the voltage variation of the thyristors is sharper than that of the load side, which causes more useful energy to be consumed by the snubber resistor due to the snubber capacitor presenting a low resistance at a high-frequency range. Reference [20] proposed that an impedance matching method can be used to protect the semiconductor devices from being damaged, which requires that the ratio of the internal resistance of the load to its inductance is far greater than that of the inductance of the pulse-shaping inductor to its internal resistance. In practical application, it is difficult to meet this requirement because the load always changes under different working conditions but the pulse-shaping inductor stays the same.

An improved PPS topology is developed in this paper. The improved PPS topology applies coupling inductor and RC snubber circuit techniques, which consumes less useful energy compared with conventional approaches because it is less sensitive to the voltage variation of the thyristors and more sensitive to the voltage variation of the load side. Besides, it also reduces the reverse recovery currents in the thyristors by absorbing the surge energy from the load side when load mutation occurs. Furthermore, the size of the pulse-shaping inductor can be reduced due to the coupling technique being applied. Finally, the simulation technique has been applied to validate theoretical analysis and the proposed model.

## 2. Reverse Recovery Current Models of the Thyristors in PPS

### 2.1. Description of PPS

The PPS is composed of multiple PFUs in parallel. Type I PFU topology and its photograph are shown in Figure 1. In Figure 1, the capacitance of the energy-storage capacitor Ci is $C_i$ ($i = 1, 2, \ldots, n$), the inductance of the pulse-shaping inductor Li is $L_i$ ($i = 1, 2, \ldots, n$), and the resistance of the load $R_{load}$ is $R_{load}$. Generally, $C_1 = C_2 = \ldots = C_n$, $L_1 = L_2 = \ldots = L_n$, $R_{load}$ is at m$\Omega$ level, the thyristor SCRi ($i = 1, 2, \ldots, n$) is used as a switch, and the fast recovery diode Di ($i = 1, 2, \ldots, n$) provides an after-flow path for the pulse-shaping inductor Li. If a PFU is discharging and the fast recovery diode is in a cut-off state, the current equation can be described as [21]:

$$L_i \frac{di(t)}{dt} + \frac{1}{C_i} \int i(t)dt + R_{load} i(t) = 0, \tag{1}$$

where $i(t)$ represents the current in the circuit.

The initial conditions are as follows:

$$\begin{cases} i(t)\big|_{t=0} = 0 \\ \frac{di(t)}{dt}\big|_{t=0} = \frac{U_0}{L_i} \end{cases}, \tag{2}$$

where $U_0$ is the initial voltage of the energy-storage capacitor.

Typically, $R_{load}$ is in the range of 5~50 m$\Omega$, $L_i$ is in the range of 5~40 $\mu$H, and $C_i$ is in the range of 1~6 mF, indicating that $\frac{R_{load}^2}{L_i^2} - \frac{4}{L_i C_i} < 0$. The angular frequency is $\omega_0 = 1/\sqrt{L_i C_i}$ and the attenuation factor is $\alpha = R_{load}/(2L_i)$. Based on Equations (1) and (2), it can be deduced:

$$i(t) = \frac{U_0}{L_i \sqrt{\omega_0^2 - \alpha^2}} e^{-\alpha t} \sin(\sqrt{\omega_0^2 - \alpha^2} t). \tag{3}$$

In practical application, the discharge time of each PFU is not synchronous [6,20]. The discharge time of PFU1 is $t_1$, the discharge time of PFU2 is $t_2$, and the discharge time of PFUn is $t_n$. Therefore, there are $n - 1$ discharge time intervals: $t_{s1} = t_2 - t_1$; $t_{s2} = t_3 - t_2$; $\ldots$; $t_{sn-1} = t_n - t_{n-1}$. The current in the load side is a superposition of the currents in the discharging PFUs.

### 2.2. The Reverse Recovery Current Models of the Thyristors in PPS

The equivalent type I PFU topology and the reverse recovery current models of the thyristors are shown in Figure 2. Figure 2a shows that if the PPS is operating, a PFU sustains backward voltages, equivalent to a reverse voltage source $V_P(t)$ and $R_{load}$ is regarded as its internal resistance, from other discharging PFUs. $L_P$ is an equivalent parasitic inductance in the energy-storage capacitor and lead wires, and its typical value is in the range of 0.05~0.2 $\mu$H. Figure 2b,c show two models of the reverse recovery currents in the thyristors in the type I PFU topology.

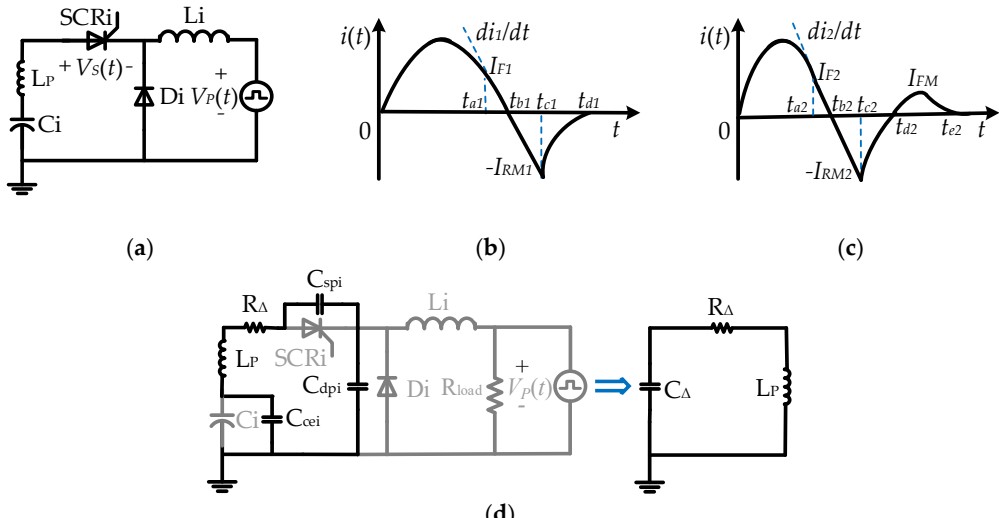

**Figure 2.** Equivalent type I PFU topology and reverse recovery current models of the thyristors. (**a**) Equivalent type I PFU topology; (**b**) reverse recovery current model No. 1 of the thyristor; (**c**) reverse recovery current model No. 2 of the thyristor; (**d**) equivalent circuit of the reverse recovery current model No. 2 when forward oscillation current occurs.

The current model No. 1 is suitable for the situation where the fast recovery diodes are not turned on in the reverse recovery phases of the thyristors. The reverse recovery current loop is mainly composed of: thyristor, energy-storage capacitor, load, and pulse-shaping inductor. When $0 < t < t_{a1}$, the energy-storage capacitor is in a discharge phase, and the current $i_1(t)$ in the thyristor is positive; when $t = t_{a1}$, $i_1(t)$ begins to decrease with a slope of $di_1/dt = -V_P(t)/L_i$ (in practical application, $L_i >> 10L_P$, thus $L_p$ is relatively small compared with $L_i$ and can be ignored), and $i_1(t_{a1}) = I_{F1}$; when $t = t_{b1}$, $i_1(t_{b1}) = 0$; when $t_{b1} < t \le t_{c1}$, $i_1(t)$ increases inversely with the same slope of $di_1/dt$ until it reaches the maximum reverse value $-I_{RM1}$; when $t_{c1} < t \le t_{d1}$, $i_1(t)$ decreases in the form of exponential function until it returns to zero at $t = t_{d1}$. The reverse recovery time can be written as: $t_{rr1} = t_{d1} - t_{b1}$.

The current model No. 2 is suitable for the situation that the fast recovery diodes have been turned on in the reverse recovery phases of the thyristors. There are two reverse recovery current loops: thyristor, energy-storage capacitor, load, and pulse-shaping inductor; thyristor, energy-storage capacitor, and fast recovery diode. When $t = t_{a2}$, the fast recovery diode has been turned on, and $i_2(t)$ begins to decrease with a slope of $di_2/dt = -[V_P(t)/L_i + V_S(t)/L_P]$; when $t = t_{b2}$, $i_2(t)$ decreases to zero; when $t = t_{c2}$, $i_2(t)$ reaches the maximum reverse value $-I_{RM2}$; when $t_{b2} < t \le t_{d2}$, the current change law is similar to that in Figure 2b; when $t_{d2} < t \le t_{e2}$, a forward oscillation current occurs due to RLC series resonance caused by parasitic inductance, capacitance, and resistance. The equivalent circuit of the reverse recovery current model No. 2 when forward oscillation current occurs is shown in Figure 2d. Due to higher reverse recovery current change rate in the current model No. 2, effects of parasitics are more obvious and the load side presents an open circuit at this moment. $C_{spi}$ represents parasitic capacitance in the thyristor SCRi, $C_{dpi}$ represents parasitic capacitance in the fast recovery diode Di, $C_{cei}$ represents equivalent capacitance of the energy-storage capacitor Ci, $R_\Delta$ represents the total parasitic resistance, and $C_\Delta$ represents the total parasitic capacitance. $R_\Delta$ is far less than $R_{load}$, and $C_\Delta = \frac{C_{spi}C_{cei}C_{dpi}}{C_{spi}C_{cei}+C_{cei}C_{dpi}+C_{dpi}C_{spi}}$.

The reverse recovery current in the thyristor shown in Figure 2b can be written as [22]:

$$i_1(t) = \begin{cases} I_{F1} - (t - t_{a1})V_P(t)/L_i & t_{a1} < t \le t_{c1} \\ -I_{RM1}\exp(-\frac{t-t_{c1}}{\tau_1}) & t_{c1} < t \end{cases}, \tag{4}$$

where $\tau_1$ is the minority carrier lifetime in the base region of the thyristor.

The relationship between the current $i_1(t)$ and the charge $Q(t)$ stored in the $\text{N}^-$ base region of a thyristor can be written as:

$$i_1(t) = \frac{dQ(t)}{dt} + \frac{Q(t)}{\tau_1}. \tag{5}$$

Based on Equations (3) and (5), it can be deduced:

$$Q(t) = \frac{GU_0}{L_i \omega^2 (1 + b^2)} (e^{-t/\tau_1} + be^{-\alpha t} \sin \omega t + e^{-\alpha t} \cos \omega t), \tag{6}$$

where $G$ is the equivalent current gain of the thyristor and $b = \frac{1 - \alpha \tau_1}{\omega \tau_1}$.

$Q(t)$ equals zero at $t_{c1}$ and $t_{s1} = t_{c1} - t_{b1}$, then $t_{c1} = t_{s1} + \pi/\omega$. Since $\omega t_{s1} << \pi/2$ and $t_{c1} >> \tau_1$, Equation (7) is deduced:

$$\tau_1 = \frac{t_{s1}}{1 + \alpha t_{s1}}. \tag{7}$$

The current model of the thyristor shown in Figure 2c is described as:

$$i_2(t) = \begin{cases} I_{F2} - (t - t_{a2})[V_P(t)/L_i + V_S(t)/L_P] & t_{a2} < t \le t_{c2} \\ -I_{RM2} \exp(-\frac{t - t_{c2}}{\tau_2}) & t_{c2} < t \le t_{d2}, \\ 2E \sqrt{C_\Delta/(4L_P - R_\Delta^2 C_\Delta)} \exp[-R_\Delta(t - t_{d2})/(2L_P)] \sin \omega_\Delta(t - t_{d2}) & t_{d2} < t \le t_{e2} \end{cases} \tag{8}$$

where $E$ is the initial voltage of the total parasitic capacitance $C_\Delta$; $\tau_2$ is the minority carrier lifetime of the base region of the thyristor; angular frequency is $\omega_\Delta = \frac{1}{\sqrt{L_P C_\Delta}} \sqrt{1 - \frac{R_\Delta^2 C_\Delta}{4 L_P}}$.

Generally, the breakdown risk increases with the reverse recovery current in a thyristor, so it is reasonable to investigate the reverse recovery current characteristic of the thyristor. The relationship between the breakdown risk and the reverse recovery current characteristic of the thyristor will be discussed in Section 2.3. Which case is suitable for the reverse recovery current model No. 1 or No. 2 of the thyristor will be identified in Section 3.

### 2.3. Breakdown Risks of the Thyristors in PPS

If initial conditions and devices' parameters keep the same, assuming that: in current model No. 1, the instantaneous voltage of a thyristor is $V_{S1}$ and its instantaneous power is $P_{SCR1}$ when $i_1(t)$ reaches its maximum reverse value $-I_{RM1}$; in current model No. 2, the instantaneous voltage of a thyristor is $V_{S2}$ and its instantaneous power is $P_{SCR2}$ when $i_2(t)$ reaches its maximum reverse value $-I_{RM2}$.

Generally, the instantaneous voltage of a thyristor approximately equals to its turn-on voltage when the current reaches its maximum reverse value, therefore, $V_{S1} = V_{S2}$. If $I_{RM1} << I_{RM2}$, $P_{SCR2} = V_{S2} I_{RM2} >> P_{SCR1} = V_{S1} I_{RM1}$, indicating that the instantaneous power of a thyristor with current model No. 2 is higher than that with current model No. 1. Therefore, the breakdown risk of the thyristor with current model No. 2 is greater than that with current model No. 1 under the same working conditions.

Documents [17,20,23] indicate that the thyristors are prone to be breakdown by sustaining overvoltages in the reverse recovery phases. Therefore, it is meaningful to investigate the factors affecting the reverse recovery currents in the thyristors and find approaches to reduce their peak values, which will be explored in Section 3.

### 3. Factors Affecting the Reverse Recovery Currents in Thyristors and Improvement of PPS Topology

If the breakdown of a thyristor occurs, the general way is to remove and replace it with a new one, which spends massive manpower and financial resources in a long term. Due to the randomness of load mutation, a typical phenomenon in a multiple uninterruptible discharge process is that the thyristors in conventional PPS topologies are not always but randomly damaged especially in the case of numerous PFUs operating together with discharge time intervals, although all initial conditions and devices' parameters keep the same. Load mutation can lead to an open circuit or sharp increase in

resistance of the load side, which results in instantaneous overvoltage and high current change rate *di/dt* in pulse-shaping inductors. The instantaneous overvoltages increase reverse recovery currents in the thyristors and cause breakdowns of them. Breakdowns of the thyristors often occur in their reverse recovery phases, which are also related to the factors such as the initial voltage of the energy-storage capacitor, the discharge time intervals, and the load resistance.

Figure 3 shows two simplified PFU topologies. Figure 3a shows the type I PFU topology and Figure 3b shows the type II PFU topology. It is displayed that the difference in the two types of PFU topologies is the locations of the fast recovery diodes.

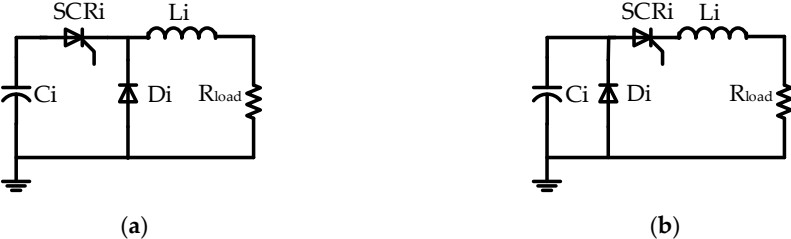

(**a**)　　　　　　　　　　　　　　　　　　(**b**)

**Figure 3.** Two simplified PFU topologies. (**a**) Simplified type I PFU topology; (**b**) simplified type II PFU topology.

### 3.1. Validation of the Two Current Models of Thyristors

To validate the circuit models No. 1 and No. 2 of thyristors, the simulation circuit of type I PPS is established in software Simplorer as shown in Figure 4. $ESL_i$ represents the series parasitic inductance of the energy-storage capacitor Ci, $ESR_i$ represents the series parasitic resistance of the energy-storage capacitor Ci, $R_{Li}$ represents the series parasitic resistance of the pulse-shaping inductor Li. The devices' parameters of PPS are shown in Table 1, and *n* is the number of PFUs.

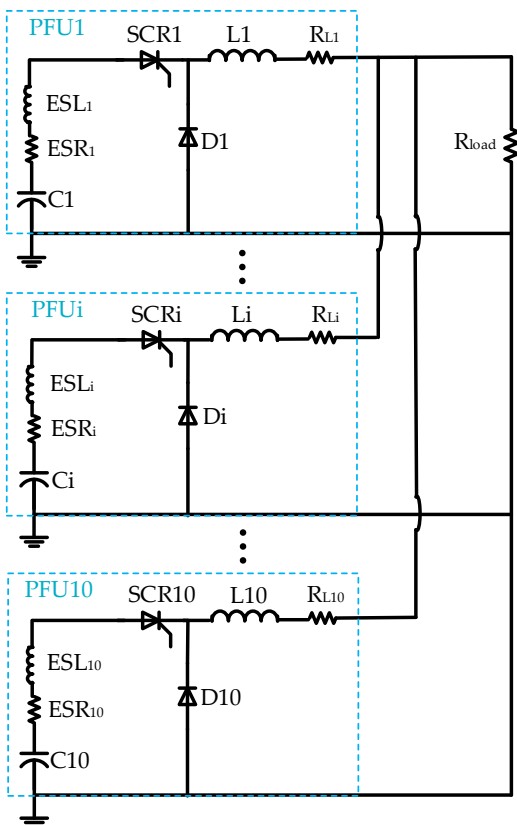

**Figure 4.** Simulation circuit of type I PPS.

**Table 1.** Devices' parameters of PPS.

| $C_i$ (mF) | $ESL_i$ (μH) | $ESR_i$ (mΩ) | $L_i$ (μH) | $R_{Li}$ (mΩ) | $n$ |
|---|---|---|---|---|---|
| 1 | 0.2 | 2 | 5 | 0.5 | 10 |

Figure 5 shows the simulation current curves of the thyristors and fast recovery diodes in type I PPS topology. The initial voltage of the energy-storage capacitor $U_0$ = 5 kV, the discharge time intervals $t_{s1} = t_{s2} = \ldots = t_{sn-1} = t_{sn} = t_s = 100$ μs, the load resistance $R_{load}$ = 50 mΩ, and other devices' parameters keep the same as those in Table 1. SCR1 represents the current curves of the thyristor SCR1, $\ldots$ , SCR10 represents the current curves of the thyristor SCR10, and D1 represents the current curves of the fast recovery diode D1, $\ldots$ , D10 represents the current curves of the fast recovery diode D10.

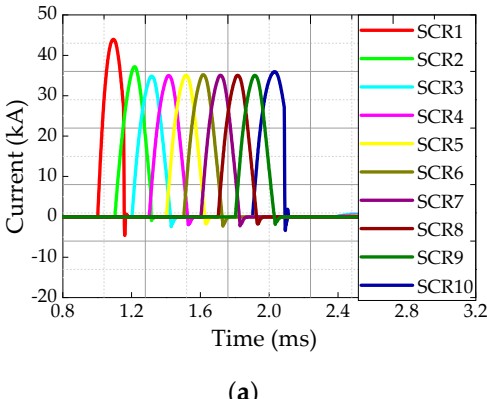
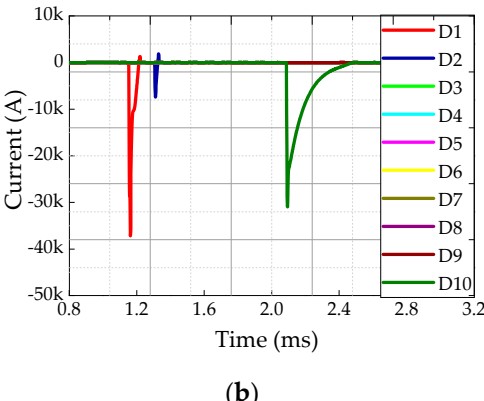

(a)　　　　　　　　　　　　　　　　　(b)

**Figure 5.** Simulation current curves of the thyristors and fast recovery diodes in type I PPS topology. (**a**) Current curves of the thyristors; (**b**) current curves of the fast recovery diodes.

The two current models of thyristors mentioned in Section 2.2 can be validated in Figure 5a. It is displayed in Figure 5b that the currents of the fast recovery diodes D3–D9 are zero, indicating that they are not turned on; the current of the fast recovery diode D2 is relatively small, indicating it is slightly turned on; the currents of the fast recovery diodes D1 and D10 are relatively large, indicating that they are fully turned on. Based on Figure 5a,b, it is found that if the peak values of the reverse recovery currents in the thyristors SCR1 and SCR10 are greater and their reverse recovery time values are shorter than those of the others due to the fast recovery diodes D1 and D10 being fully turned on. It is revealed that there are forward oscillation currents in the thyristors SCR1 and SCR10, thus the current models No. 1 and No. 2 of the thyristors in PPS are validated. It is also exhibited that if the fast recovery diodes, especially D1 and D10, are turned on in type I PPS topology, the peak values of the reverse recovery currents in the thyristors are increased, which results in appearance of the forward oscillation currents and increases the breakdown risks of the thyristors.

*3.2. Effects of the Initial Voltage of the Energy-Storage Capacitor on the Reverse Recovery Currents in Thyristors*

Figure 6 shows the simulation current curves of the thyristors in the type I PPS topology with different initial voltages of the energy-storage capacitor based on the simulation circuit shown in Figure 4. The initial conditions and devices' parameters keep the same as those in Section 3.1, except that the initial voltage of the energy-storage capacitor varies. Since there are two thyristor current models in the type I PPS topology, the thyristors SCR1 and SCR6 are selected as the research objects to observe the effects of the initial voltage of the energy-storage capacitor on the reverse recovery currents. The thyristor SCR1 and the fast recovery diode D1 are in PFU1. The fast recovery diode D1 is turned on under this condition. The thyristor SCR6 and the fast recovery diode D6 are in PFU6. The fast recovery diode D6 is not turned on under this condition.

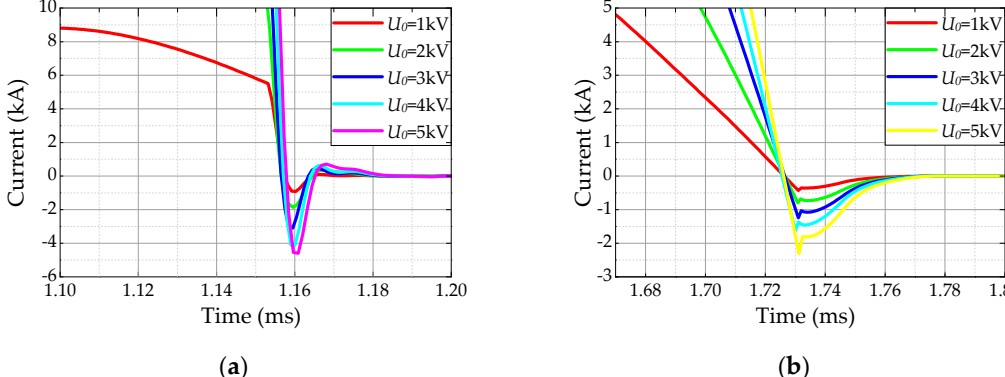

**Figure 6.** Simulation current curves of the thyristors in the type I PPS topology with different initial voltages of the energy-storage capacitor. (**a**) Current curves of the thyristor SCR1; (**b**) current curves of the thyristor SCR6.

Figure 6a illustrates that the peak value of the forward oscillation current in the thyristor SCR1 increases with the increase of the initial voltage of the energy-storage capacitor. It is also revealed that if the initial voltage of the energy-storage capacitor increases, the peak value of the reverse recovery current in the thyristor SCR1 and its reverse recovery time increase. Besides, the peak value of the reverse recovery current in the thyristor SCR1 is larger and its reverse recovery time is shorter than those of the thyristor SCR6. It can be inferred that the smaller the initial voltage of the energy-storage capacitor is, the less likely the thyristor will be damaged.

### 3.3. Effects of the Discharge Time Intervals on the Reverse Recovery Currents in Thyristors

Figure 7 shows the simulation current curves of the thyristors and voltage curves of the load side in the type I PPS topology with different discharge time intervals. The initial conditions and devices' parameters keep the same as those in Section 3.1 except that the discharge time intervals vary. The thyristors SCR1 and SCR6 are also selected as the research objects.

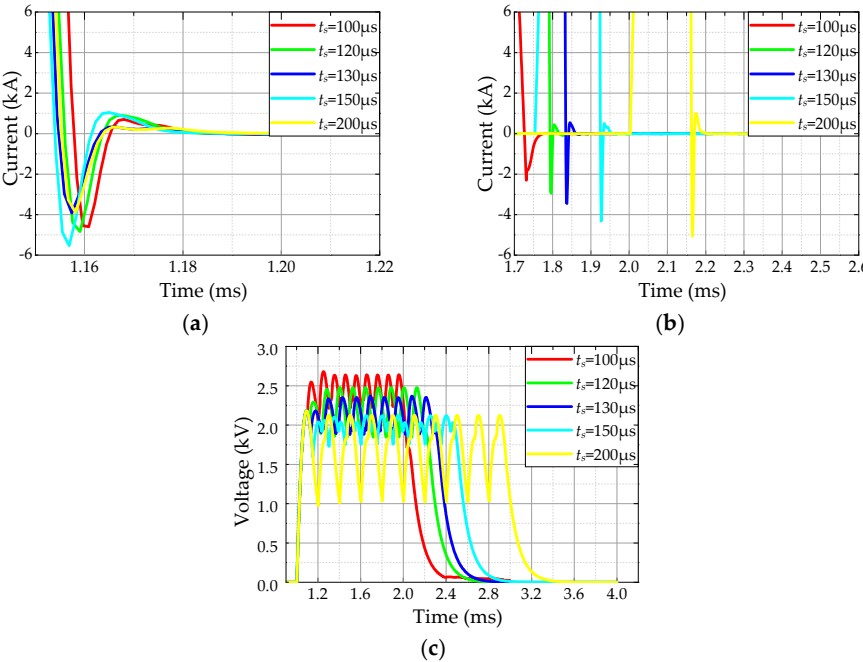

**Figure 7.** Simulation current curves of the thyristors and voltage curves of the load side in the type I PPS topology with different discharge time intervals. (**a**) Current curves of the thyristor SCR1; (**b**) current curves of the thyristor SCR6; (**c**) voltage curves of the load side.

Figure 7a illustrates that the peak values of the reverse recovery and forward oscillation currents, and its reverse recovery time of the thyristor SCR1 do not increase but vibrate with the increase of the discharge time intervals. Figure 7b illustrates that the peak values of the reverse recovery and forward oscillation currents in the thyristor SCR6 increase with the increase of the discharge time intervals. It is also displayed that the forward oscillation current gradually occurs with the increase of the discharge time intervals, indicating that the fast recovery diode D6 is gradually turned on. Figure 7c illustrates that the voltage amplitude of the load side decreases and its ripple increases with the increase of the discharge time intervals, which indicates that the reverse voltage source $V_P(t)$ sustained by the thyristor is not stable, resulting in the phenomenon shown in Figure 7a.

It can be inferred that with the increase of the discharge time intervals, the number of the turned-on fast recovery diodes increases, and the number of the thyristors with forward oscillation currents increases, but the amplitude of $V_P(t)$ reduces and tends to a stable amplitude finally, which indicate that the breakdown risks of the thyristors increase, and a similar result is given in [20]. It should be noted that with the increase of the discharge time intervals to a certain extent, $V_P(t)$ is no longer a flat-topped wave, which will fail to meet the working requirements of the load side in practical application.

### 3.4. Effects of the Load Resistance on the Reverse Recovery Currents in Thyristors

Figure 8 shows the simulation current curves of the thyristors in the type I PPS topology with different load resistances. The initial conditions and devices' parameters keep the same as those in Section 3.1 except that the load resistance varies. The thyristors SCR1 and SCR6 are also selected as the research object.

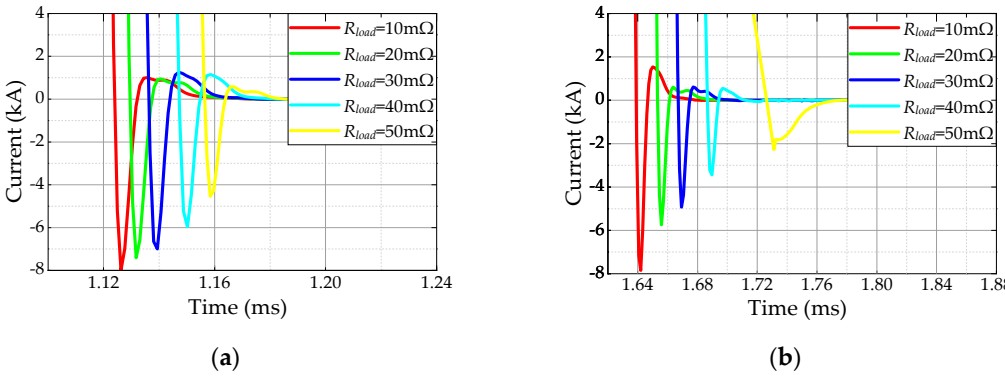

(**a**)          (**b**)

**Figure 8.** Simulation current curves of the thyristors in the type I PPS topology with different load resistances. (**a**) Current curves of the thyristor SCR1; (**b**) current curves of the thyristor SCR6.

Figure 8a,b illustrates that with the increase of load resistance, there is a decrease in the peak values of the reverse recovery currents in the thyristors SCR1 and SCR6. Similarly, their forward oscillation currents exhibit reduction in peak values, and the moments of entering the reverse recovery phases are delayed. In Figure 8b, the forward oscillation current in the thyristor SCR6 even disappears when $R_{load} = 50$ mΩ. It can be inferred that with the increase of the load resistance, the number of the thyristors with forward oscillation currents decreases. Therefore, the breakdown risks of the thyristors in PPS can be reduced by increasing the load resistance.

### 3.5. The Reverse Recovery Currents in the Thyristors in Type II PPS Topology

The simulation circuit of type II PPS is shown in Figure 9 and the simulation results are shown in Figure 10. Figure 10a shows the simulation current curves of the thyristors in the type II PPS topology, and its initial conditions and devices' parameters keep the same as those in Section 3.1. Figure 10a illustrates that the peak values of the reverse recovery currents in all thyristors are almost consistent in the type II PPS topology. Similarly, their reverse recovery time values are almost consistent, and there are no forward oscillation currents. Comparing Figure 10a,b, it is found that no matter whether the

fast recovery diodes are turned on or not, there is no increase in the peak values of the reverse recovery currents and appearance of the forward oscillation currents in the thyristors in the type II PPS topology. Comparing Figures 5a and 10a, it is found that the peak values of the reverse recovery currents in the thyristors, especially SCR1 and SCR 10 in the type II PPS topology, are lower than those in the type I PPS topology.

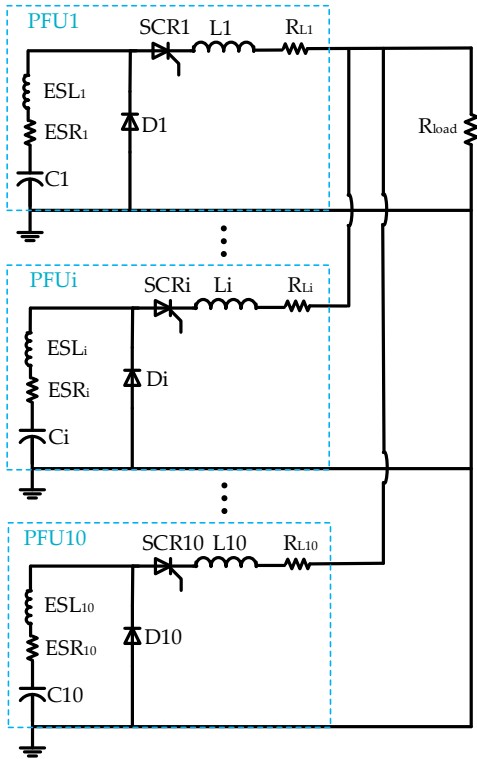

**Figure 9.** Simulation circuit of type II PPS.

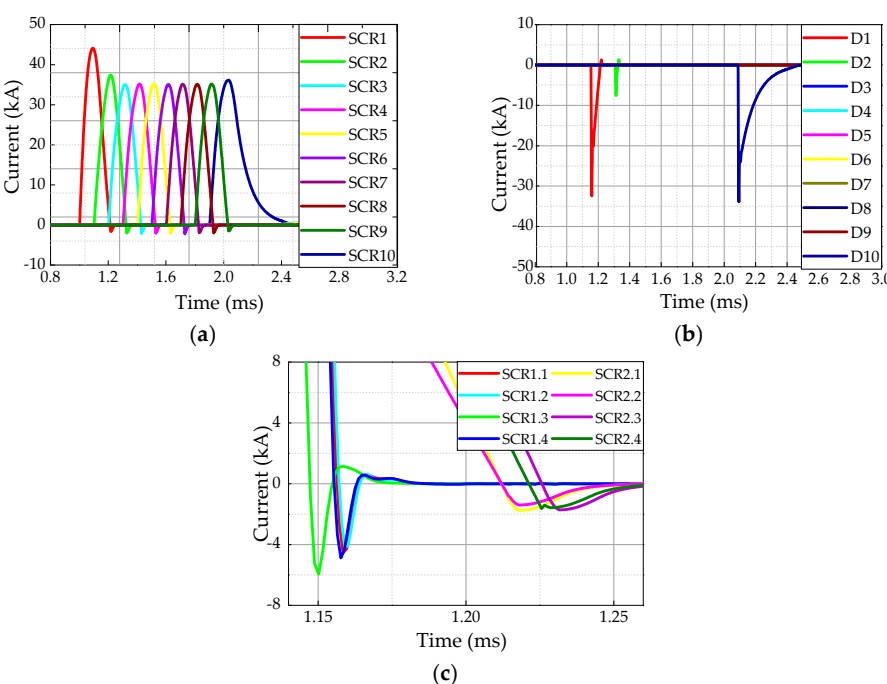

**Figure 10.** Simulation current curves of the thyristors and fast recovery diodes in conventional PPS. (**a**) Current curves of the thyristors in the type II PPS topology; (**b**) current curves of the fast recovery diodes in the type II PPS topology; (**c**) current curves of the thyristor SCR1 in type I and II PPS topologies.

To analyze the reverse recovery currents in the thyristors in type I and II PPS topologies, the thyristor SCR1 in PFU1 is selected as a research object, and the simulation results are shown in Figure 10c. SCR 1.1 represents the simulation current curve of the thyristor SCR1 in the type I PPS topology when the initial voltage of the energy-storage capacitor $U_0 = 5$ kV, the discharge time intervals $t_r = 100$ µs, and the load resistance $R_{load} = 50$ mΩ. SCR 1.2 represents the simulation current curve of the thyristor SCR1 in the type I PPS topology when the initial voltage of the energy-storage capacitor $U_0 = 4$ kV, the discharge time intervals $t_r = 100$ µs, and the load resistance $R_{load} = 50$ mΩ. SCR 1.3 represents the simulation current curve of the thyristor SCR1 in the type I PPS topology when the initial voltage of the energy-storage capacitor $U_0 = 5$ kV, the discharge time intervals $t_r = 100$ µs, and the load resistance $R_{load} = 40$ mΩ. SCR 1.4 represents the simulation current curve of the thyristor SCR1 in the type I PPS topology when the initial voltage of the energy-storage capacitor $U_0 = 5$ kV, the discharge time intervals $t_r = 120$ µs, and the load resistance $R_{load} = 50$ mΩ. SCR 2.1 represents the simulation current curve of the thyristor SCR1 in the type II PPS topology when the initial voltage of the energy-storage capacitor $U_0 = 5$ kV, the discharge time intervals $t_r = 100$ µs, and the load resistance $R_{load} = 50$ mΩ. SCR 2.2 represents the simulation current curve of the thyristor SCR1 in the type II PPS topology when the initial voltage of the energy-storage capacitor $U_0 = 4$ kV, the discharge time intervals $t_r = 100$ µs, and the load resistance $R_{load} = 50$ mΩ. SCR 2.3 represents the simulation current curve of the thyristor SCR1 in the type II PPS topology when the initial voltage of the energy-storage capacitor $U_0 = 5$ kV, the discharge time intervals $t_r = 100$ µs, and the load resistance $R_{load} = 40$ mΩ. SCR 2.4 represents the simulation current curve of the thyristor SCR1 in the type II PPS topology when the initial voltage of the energy-storage capacitor $U_0 = 5$ kV, the discharge time intervals $t_r = 120$ µs, and the load resistance $R_{load} = 50$ mΩ.

It can be revealed that the peak values of the reverse recovery currents in the thyristor SCR1 in the type II PPS topology are smaller than those in the type I PPS topology under the same working conditions, and no forward oscillation currents in the thyristor SCR1 in the type II PPS topology are found. Comparing Figures 5 and 10, the peak values of the reverse recovery and forward oscillation currents in the thyristors in the type I PPS topology are also influenced by the fast recovery diodes. However, the influence of the fast recovery diodes on the peak values of the reverse recovery currents in the thyristors in the type II PPS topology can be ignored, and the forward oscillation currents are even not influenced by the fast recovery diodes.

It can be concluded that the influence of the initial voltage of the energy-storage capacitor, the discharge time intervals, and the load resistance on the reverse recovery currents in the thyristors in the type II PPS topology is smaller than that in the type I PPS topology, thus the breakdown risks of the thyristors can be reduced.

### 3.6. Improvement of PPS Topology

The breakdown risks of the thyristors in both type I and II PPS topologies, which are caused by the instantaneous overvoltages from the pulse-shaping inductors of other discharging PFUs due to load mutations, are relatively high. The load mutation occurs in the EML experiment due to the armature is out of or broken in the railgun barrel, which leads to an open circuit or sharp increase in resistance of the load side. The load mutation can cause high current change rates, which lead to instantaneous overvoltages of the pulse-shaping inductors. The instantaneous overvoltages can result in high reverse recovery currents in the thyristors, and cause damages of them in conventional topologies.

Based on the simulation circuits of PPS shown in Figures 4 and 9, it is found that the thyristors in both type I and II PPS topologies easily suffer from the instantaneous overvoltages from the pulse-shaping inductors. Therefore, it is meaningful to explore approaches to protect the thyristors from being damaged.

The common method is to increase the number of thyristors in series. It is a practicable way, but requires higher output power and increases the complexity of trigger circuits. Furthermore, due to the amplitudes of the instantaneous overvoltages from the pulse-shaping inductors being different in

different working conditions, which results in different amplitudes of the reverse recovery currents in the thyristors, it is not easy to determine the number of the thyristors in series, especially when the cost is considered.

To overcome the shortcomings of the conventional PPS topologies, a new PPS topology is developed. By applying the coupling technique, the pulse-shaping inductor is designed as a coupling inductor. Then, an RC snubber circuit is added between the common connection point of the two windings of the coupling inductor and the ground, which absorbs the surge energy from the load side. Based on the decoupling theory, the simplified model of the coupling inductor in PFUi is shown in Figure 11. If the inductance of one winding is $L_{ai}$, the other is $L_{bi}$, and their mutual inductance is $M_i$, then a negative inductance with value $-M_i$ is formed between the common connection point of the two windings and the RC snubber circuit. The negative inductance can be equivalent to a capacitance with value $C_{eqi}$, which can be deduced as:

$$-\mathrm{j}\omega M_i = \frac{1}{\mathrm{j}\omega C_{eqi}},$$ (9)

where $\omega$ represents angular frequency.

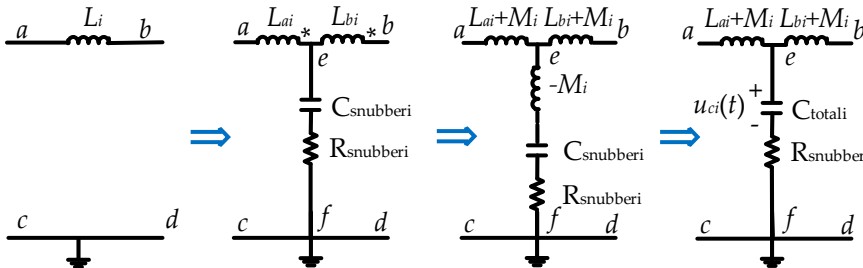

**Figure 11.** Simplified model of the coupling inductor in PFUi.

Based on (9), the equivalent capacitance is written as: $C_{eqi} = \frac{1}{\omega^2 M_i}$, thus the total capacitance $C_{totali}$ of the ef branch is:

$$C_{totali} = \frac{C_{eqi} C_{snubberi}}{C_{eqi} + C_{snubberi}}.$$ (10)

If the voltage of the equivalent capacitance $C_{totali}$ is $U_{ci}(t)$, then the energy absorbed by the ef branch from $t_0$ to $t$ is:

$$W(t) = \frac{1}{2} C_{totali} [u_{ci}{}^2(t) - u_{ci}{}^2(t_0)] + C_{totali}(R_{snubberi} + R_{ci}) \int_{t_0}^{t} [\frac{du_{ci}(\tau)}{d\tau}]^2 d\tau,$$ (11)

where $R_{ci}$ is the parasitic resistance of the equivalent capacitance $C_{totali}$, and $R_{snubberi}$ is the resistance of the snubber resistor $R_{snubberi}$.

Due to the ef branch consuming a relatively small part of the useful energy, it can be ignored when PFUi is operating and there is:

$$L_i = L_{ai} + L_{bi} + 2M_i.$$ (12)

Figure 12 shows the simulation circuit of the improved PPS. The initial conditions and devices' parameters keep the same as those in Section 3.1, $L_{ai} = L_{bi} = 2\ \mu H$, coupling factor of the inductor $k = 0.25$, $C_{snubberi} = 10\ \mu F$, and $R_{snubberi} = 30\ \Omega$. The switch S is open at $t = 2$ ms, resulting in an open circuit at the load side.

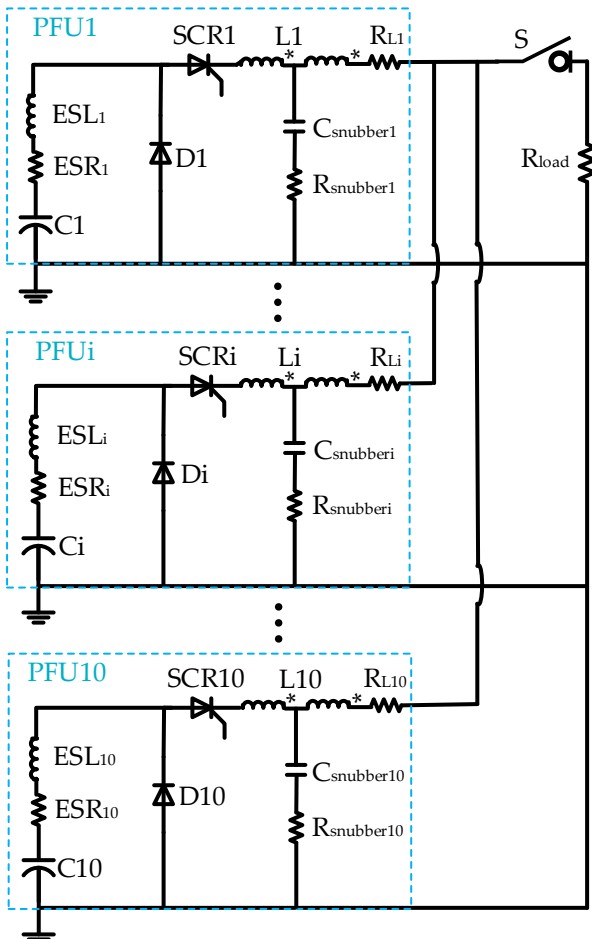

**Figure 12.** Simulation circuit of improved PPS.

The simulation voltage curves of pulse-shaping inductors with high voltage spikes in the type II PPS topology are shown in Figure 13a. It is displayed that the high voltage spike of the pulse-shaping inductor L9 is formed in the after-flow phase when load mutation occurs, while the high voltage spike of the pulse-shaping inductor L10 is formed before its after-flow phase. It is also exhibited that the value of the high voltage spike of the pulse-shaping inductor L10 is greater than that of the pulse-shaping inductor L9, which will inevitably cause the thyristor SCR9 to suffer from the high voltage spike of the pulse-shaping inductor L10.

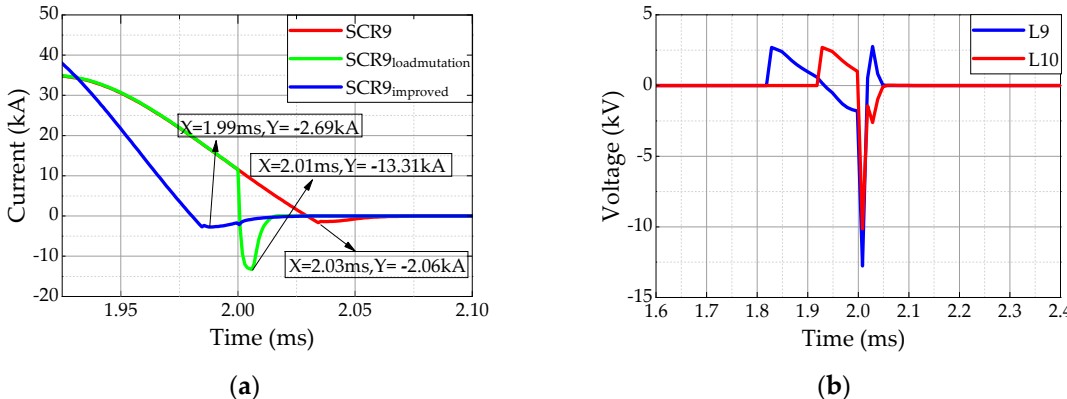

**Figure 13.** Simulation current curves of the thyristor SCR9 and voltage curves of the pulse-shaping inductors L9 and L10. (**a**) Voltage curves of the pulse-shaping inductors L9 and L10 in type II PPS topology; (**b**) current curves of the thyristor SCR9 in type II and improved PPS topologies.

The simulation current curves of the thyristor SCR9 is shown in Figure 13b. In Figure 13b, SCR9 represents the current curve of the thyristor SCR9 in the type II PPS topology without load mutation, SCR$_{\mathrm{loadmutation}}$ represents the current curve of the thyristor SCR9 in the type II PPS topology with load mutation, and SCR$_{\mathrm{improved}}$ represents the current curve of the thyristor SCR9 in the improved PPS topology with load mutation.

Figure 13b illustrates that the peak value of the reverse recovery current in the thyristor SCR9 in the type II PPS topology with load mutation at about 5 times larger than that without load mutation, and the peak value of the reverse recovery current in the thyristor SCR9 in improved topology with load mutation is almost equal to that in the type II PPS topology without load mutation. Therefore, it is reasonable to apply the improved PPS topology to reduce the breakdown risks of the thyristors and improve the reliability of conventional PPS.

To sum up, the thyristors in conventional PPS topologies have larger peak values of the reverse recovery currents if sustaining instantaneous overvoltages from the pulse-shaping inductors of other discharging PFUs caused by load mutation in the reverse recovery phases, while the thyristors in the improved PPS topology have lower peak values of the reverse recovery currents due to the RC snubber circuits can absorb the surge energy from the load side when load mutation occurs.

### 3.7. Test Results of EML Experiments

The test layout of the EML experiment is shown in Figure 14. In the experiment, the load is an electromagnetic railgun, which is mainly composed of a copper barrel and an aluminum armature. The resistance of the electromagnetic railgun is very small, which is at the m$\Omega$ level. The control system sends charging signals to the chargers initially; then, the chargers provide energy for the energy-storage capacitors; next, when the preset voltage of the energy-storage capacitors is reached, the control system sends charging-stop signals to the chargers, and the charging process is over; ultimately, when the PPS receives a firing signal from the control system, the PFUs discharge according to the preset discharge time intervals. The currents in the thyristors can be measured by Rogowski current probes which can send their collected results to the data analysis system. The data analysis system can automatically analyze the measured current data and present it for the testers.

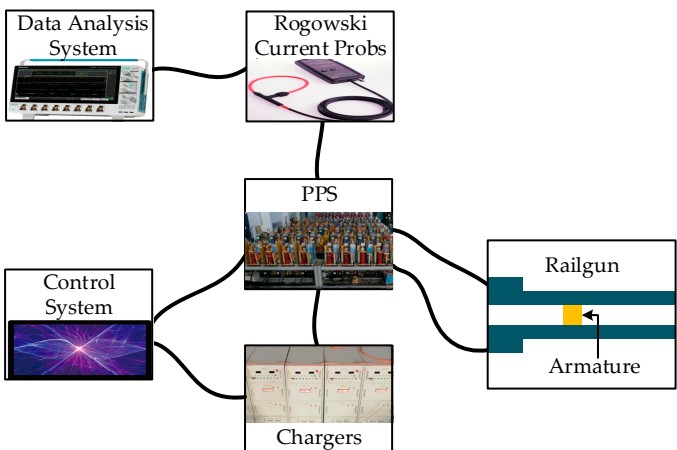

**Figure 14.** Test layout of the EML experiments.

The measured current curves of the thyristors in type I PPS topology are shown in Figure 15. Figure 15a shows the measured current curve of a thyristor in the type I PPS topology when the initial voltage of the energy-storage capacitor $U_0 = 5$ kV and the discharge time intervals $t_r = 200$ μs. It is observed that the thyristor in Figure 15a has a reverse recovery current with the peak value $-3.93$ kA. The thyristor has not been damaged in this case. Figure 15b shows the measured current curve of a thyristor in the type I PPS topology when the initial voltage of the energy-storage capacitor $U_0 = 8$ kV and the discharge time intervals $t_r = 200$ μs. It found that the armature is broken in the barrel and the

thyristor is damaged. Figure 15b illustrates that the damaged thyristor is in a low resistance state and loses its reverse blocking capability.

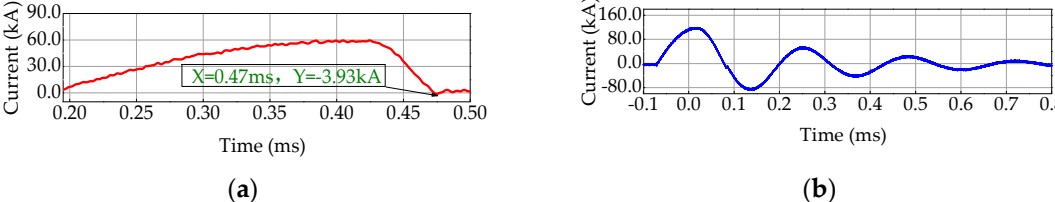

(**a**)                    (**b**)

**Figure 15.** Measured current curves of the thyristors in the type I PPS topology. (**a**) Current curve of the thyristor without being damaged; (**b**) current curve of the damaged thyristor.

The photograph of the damaged thyristor is shown in Figure 16. The maximum forward and reverse breakdown voltages of the thyristor are far greater than its working voltage. It can be inferred that the breakdown of the thyristor occurs in its reverse recovery phase. The breakdown of the thyristor is caused by the instantaneous overvoltages from the pulse-shaping inductors of other discharging PFUs. Once the breakdown of the thyristor occurs, it will suffer from a large current in a long time due to the disappearance of the reverse recovery blocking capability, which also inevitably causes damage to it. Therefore, it is meaningful for the designers of PPS to consider the load mutation and develop new topologies.

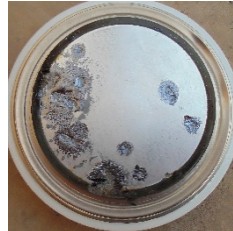

**Figure 16.** Photograph of the damaged thyristor.

## 4. Conclusions

In this paper, the reverse recovery currents in the thyristors in PPS are investigated, which is helpful to reduce the breakdown risks of the thyristors. Two cases of the reverse recovery currents are analyzed initially. Then, the effects of the initial voltage of the energy-storage capacitor, the discharge time intervals, and the load resistance on the reverse recovery currents are explored.

The research results are as follows: large initial voltage of the energy-storage capacitor leads to large reverse recovery currents, which increases breakdown risks of the thyristors; large discharge time intervals increase the number of the thyristors with forward oscillation currents, which increases the breakdown risks of the thyristors; large load resistance leads to less number of the thyristors with the forward oscillation currents, which reduces the breakdown risks of the thyristors. It is also found that the type II PPS topology is more helpful to reduce the breakdown risks of the thyristors under the same working conditions. However, the reverse recovery currents in the thyristors in both type I and II topologies are relatively high when sustaining instantaneous overvoltages caused by load mutations.

To overcome the shortcomings of the conventional PPS topologies, an improved PPS topology is proposed, which consumes less useful energy compared with conventional approaches and absorbs the surge energy from the load side more quickly when load mutation occurs. Therefore, the improved PPS topology can reduce the reverse recovery currents and breakdown risks of the thyristors.

**Author Contributions:** Conceptualization, J.W. and B.L.; methodology, J.W. and Z.L.; formal analysis, J.W., Z.L. and B.L.; investigation, J.W. and B.L.; data curation, J.W.; simulation, J.W.; validation, J.W. and Z.L.; writing—original draft preparation, J.W.; writing—review and editing, J.W., Z.L. and B.L.; supervision, Z.L. and B.L. All authors have read and agreed to the published version of the manuscript.

**Funding:** This research received no external funding.

**Conflicts of Interest:** The authors declare no conflict of interest.

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
