# Peer review of "Investigation of Reverse Recovery Current of High-Power Thyristor in Pulsed Power Supply"

_electronics, doi:10.3390/electronics9081292_

Round 1
Reviewer 1 Report
See attached file

Author Response
Dear Reviewer,
Thank you for your comments concerning our manuscript entitled “Influence of External Factors on Reverse Recovery Characteristics of High-power Thyristors in Pulsed Power Supply” (Manuscript ID: electronics-869933), which has been changed to ” Investigation of Reverse Recovery Current of High-power Thyristor in Pulsed Power Supply ”. Those comments are all valuable and very helpful for revising and improving our paper, as well as the important guiding significance to our researches. We have studied comments carefully and have made corrections which we hope meet with approval. Revised portions are marked in red in the paper. The responses to your comments are as follows:
- In this paper, the influence of external factors (initial voltage, discharge time intervals, and load resistance) on the reverse recovery characteristics of high-power thyristors in Pulsed Power Systems is investigated. Authors demonstrated that the breakdown risks of high-power thyristors in PPS increase with the reverse recovery and forward oscillation currents in thyristors. Compared with type-Ι PPS topology, the type-ΙΙ PPS topology is more helpful to limit the reverse recovery and forward oscillation currents, which is advocated to reduce breakdown risks of high-power thyristors and maintenance costs of PPS.
The language of this manuscript is understandable. The overall quality of the manuscript is well supported and documented. But, from my point of view, the technique used in, as authors named it, “Type II PFU topology” is very well known. Placing a diode in parallel with the storage capacitance stops all the reverse currents in the switching device.
Response: We have added new content: improvement of conventional PPS topology (line 308-377). But I have a question: why can we stop all the reverse currents in the switching device by placing a diode in parallel with the storage capacitance? From my point of view, the reverse current exists in every thyristor according to the theory of the reverse recovery characteristic. The reverse current can be very large or small, which depends on the working conditions. It is necessary to reduce the reverse current if it is very large especially in super high power applications, where the requirements for the manufacturing engineering of high power thyristors are higher and higher and the sizes for the thyristors are smaller and smaller.
We would be greatly appreciated if you reconsider it and make suggestions!
Attached please find the revised version, which we would like to submit for your kind consideration. We would like to express our great appreciation to you and reviewers for comments on our paper. Looking forward to hearing from you.
Once again, thank you very much for your comments and suggestions.
Best regards!
Name: Jiufu Wei
E-mail: wjfnjust@163.com
Name: Zhenxiao Li
E-mail: lizhxnjust@126.com
Name: Baoming Li (corresponding author)
Reviewer 2 Report
Dear Authors,
I read your paper, and there are my comments:
- The introduction is not acceptable. The authors should review the background and state-of-the-art of this area, not only mentioning some references.
- The equations of section 2.1 are determined by authors? if not, authors should mention the reference.
- What readers can realize in Figure 2 (b) and (c)? it needs more discussions.
- This paper investigates the impact of some voltage and current changes on two models of PFUs. And, based on the conclusion the type-II has better performance. It is essential to propose another model by authors that can improve the performance of type-II to improve the quality of the paper.
- Reference 22 should modify.
Totally, in my opinion, this paper has a good comparison between the two models, but, it is essential to propose another model to improve the existing models.
Author Response
Dear Reviewer,
Thank you for your comments concerning our manuscript entitled “Influence of External Factors on Reverse Recovery Characteristics of High-power Thyristors in Pulsed Power Supply” (Manuscript ID: electronics-869933), which has been changed to ” Investigation of Reverse Recovery Current of High-power Thyristor in Pulsed Power Supply ”. Those comments are all valuable and very helpful for revising and improving our paper, as well as the important guiding significance to our researches. We have studied comments carefully and have made corrections which we hope meet with approval. Revised portions are marked in red in the paper. The responses to your comments are as follows:
1. The introduction is not acceptable. The authors should review the background and state-of-the-art of this area, not only mentioning some references.
Response: We have reorganized the contents in introduction (line 22-75). See the article for details.
2. The equations of section 2.1 are determined by authors? if not, authors should mention the reference.
Response: It has been revised according to your opinion (line 84).
3. What readers can realize in Figure 2 (b) and (c)? it needs more discussions.
Response: Figure 2 (b) and (c) show theoretical models of the reverse recovery currents of the thyristors in PPS. The purpose of this paper is to explore the accesses to reduce the reverse recovery currents of the thyristors related to the breakdown risks and improve the reliability of PPS. The theoretical models in Figure 2 (b) and (c) are applied to evaluate the cases discussed in Section 3, and the contents presented in Section 3 can be applied to validate the theoretical models in Figure 2 (b) and (c). Therefore, the readers must combine Section 2 with Section 3.
4. This paper investigates the impact of some voltage and current changes on two models of PFUs. And, based on the conclusion the type-II has better performance. It is essential to propose another model by authors that can improve the performance of type-II to improve the quality of the paper.
Response: A improved PPS topology is developed. See Section 3.6 (line 308-377).
5. Reference 22 should modify.
Response: We are sorry for our negligence in the reference format. It has been corrected.
Attached please find the revised version, which we would like to submit for your kind consideration. We would like to express our great appreciation to you and reviewers for comments on our paper. Looking forward to hearing from you.
Once again, thank you very much for your comments and suggestions.
Best regards!
Name: Jiufu Wei
E-mail: wjfnjust@163.com
Name: Zhenxiao Li
E-mail: lizhxnjust@126.com
Name: Baoming Li (corresponding author)
Reviewer 3 Report
Authors present an analysis of electronic devices. The focus of the paper is not so clear.
Some suggestions:
Line 34-43: Please, add reference related to the sentences in this paragraph. What is ‘electromagnetic launching (EML) experiments’?
Line 93 and 100: Why authors refer about maximum for a negative quantity ‘the maximum value of −IRM1;’?
Line 134: Please, clarify the sentence ‘If the breakdown of a thyristor occurs, an RLC resonance circuit will be formed…..
Line 141 what is the sense of the sentence ‘Once PPS is made into a real object, the space occupied by thyristors is fixed. If the breakdown… ‘? What is the topology proposed to avoid the problem?’
Fig 5 refer to more devices. How they are connected? Please, add a schema
The reported results refer to experiments or simulations? Please, clarify
Line 205: Authors refer to failure. Please, add a figure where the effect of the failure is shown incurrent-time curve
Line 241 Please, support the sentence ’It can be concluded that the influence of initial voltage of energy-storage capacitor’ with results
Please, improve the aim of the paper. What is the meaning of the obtained results? What is the target? There is an experimental evidence?
What is the analysis method? Only one case? How the authors vary the studied parameter? The use a particular software for analysis? Please, describe the analysis approach. The aim is the robustness of the device? Please, add reference about this topic
Author Response
Dear Reviewer,
Thank you for your comments concerning our manuscript entitled “Influence of External Factors on Reverse Recovery Characteristics of High-power Thyristors in Pulsed Power Supply” (Manuscript ID: electronics-869933), which has been changed to ” Investigation of Reverse Recovery Current of High-power Thyristor in Pulsed Power Supply ”. Those comments are all valuable and very helpful for revising and improving our paper, as well as the important guiding significance to our researches. We have studied comments carefully and have made corrections which we hope meet with approval. Revised portions are marked in red in the paper. The responses to your comments are as follows:
1. Line 34-43: Please, add reference related to the sentences in this paragraph. What is ‘electromagnetic launching (EML) experiments’?
Response: The reference related to EML experiments have been added in the paper (line 39). (See reference [5]: “Design and Testing of a 10-MJ Electromagnetic Launch Facility”)
2. Line 93 and 100: Why authors refer about maximum for a negative quantity ‘the maximum value of −IRM1’?
Response: We are sorry for our incorrect writing. It is revised as follows: the maximum value −IRM1 (line 112).
3. Line 134: Please, clarify the sentence ‘If the breakdown of a thyristor occurs, an RLC resonance circuit will be formed…..’
Response: The components of each PFU in the PPS are: capacitor, thyristor, fast recovery diodes, inductor, and load. The load presents a characteristic of resistance in practical application, so an RLC resonance circuit in a PFU is inevitablely formed if the thyristor is breakdown, which can be validated in Figure 15 (b). A breakdown thyristor loses its reverse blocking capacity and presents a very low resistance. Theoretically, the energy is consumed only by the load and the parasitic resistance in the circuit.
4. Line 141 what is the sense of the sentence ‘Once PPS is made into a real object, the space occupied by thyristors is fixed. If the breakdown… ‘? What is the topology proposed to avoid the problem? Fig 5 refer to more devices. How they are connected? Please, add a schema. The reported results refer to experiments or simulations? Please, clarify.
Response: In recent years, the development of PPS tends to high energy storage density and small volume, so the space for the thyristors is limited in practical application. It is meaningful to enhance the robustness of PPS by developing new topology, which is presented in Section 3.6 in the revised manuscript (line 308-377). The content of “more devices” is deleted in the revised manuscript because it almost has no relevance to the object of the article. The reported results have been clarified in the revised manuscript.
5. Line 205: Authors refer to failure. Please, add a figure where the effect of the failure is shown in current-time curve.
Response: We are sorry for not finding a picture where the effect of the failure is shown in current-time curve, because we found that the initial speed of the armature in a railgun will be reduced by adopting improper discharge time intervals, although the current-time curve is normal. Therefore, it is rewritten as: “which will fail to meet the working requirements of the load side in practical application” (line 242-243 ).
6. Line 241 Please, support the sentence ’It can be concluded that the influence of initial voltage of energy-storage capacitor’ with results.
Response: Some content has been added in Section 3.5 according to your opinion (line 265-300).
Attached please find the revised version, which we would like to submit for your kind consideration. We would like to express our great appreciation to you and reviewers for comments on our paper. Looking forward to hearing from you.
Once again, thank you very much for your comments and suggestions.
Best regards!
Name: Jiufu Wei
E-mail: wjfnjust@163.com
Name: Zhenxiao Li
E-mail: lizhxnjust@126.com
Name: Baoming Li (corresponding author)
Reviewer 4 Report
This paper presents the results of a study concerning the influence of external factors over the reverse recovery characteristics of thyristors used in Pulse Power supply devices.
The results presented by the authors are interesting and the conclusions can be used for the efficient and secure design of PPS solutions. However, the reviewer recommend the authors to take into consideration the following suggestions:
- Define acronyms at their first use. For instance, define SCR in figures 1, 2 and 4.
- Page 4, line 122: check if the current used to compute power P_SCR2 is correct (i.e. you should use I_RM2 instead of I_RM1);
- Page 5, lines 154-155: check the definitions of ESL_i and ESR_i; at present they are the same.
- Figures 6 and 7: bring the caption on the same page with the figure.
- Page 8, end of subsection 3.4: considering the objective of the paper (study of the thyristors’ breakdown risk), a conclusion should be drawn on the influence of load resistance on the risk of breakdown for thyristors.
- Page 9, conclusion in lines 241-244: this conclusion is established exclusively from the analysis of the shape of thyristors’ current curves in the case of Type-II PPS topology. It would be correct to draw the conclusion by comparing curves similar to those in Figures 6, 7 and 8, but constructed for type-II PPS topology.
Author Response
Dear Reviewer,
Thank you for your comments concerning our manuscript entitled “Influence of External Factors on Reverse Recovery Characteristics of High-power Thyristors in Pulsed Power Supply” (Manuscript ID: electronics-869933), which has been changed to ” Investigation of Reverse Recovery Current of High-power Thyristor in Pulsed Power Supply ”. Those comments are all valuable and very helpful for revising and improving our paper, as well as the important guiding significance to our researches. We have studied comments carefully and have made corrections which we hope meet with approval. Revised portions are marked in red in the paper. The responses to your comments are as follows:
1. Define acronyms at their first use. For instance, define SCR in figures 1, 2 and 4.
Response: It has been revised according to your opinion.
2. Page 4, line 122: check if the current used to compute power P_SCR2 is correct (i.e. you should use I_RM2 instead of I_RM1);
Response: It is corrected: PSCR2=VS2IRM2 instead of PSCR2=VS2IRM1 (line147 ).
3. Page 5, lines 154-155: check the definitions of ESL_i and ESR_i; at present they are the same.
Response: It is corrected: ESLi represents the series parasitic inductance of the energy- storage capacitor Ci, ESRi represents the series parasitic resistance of the energy-storage capacitor Ci (line 173-176).
4. Figures 6 and 7: bring the caption on the same page with the figure.
Response: It has been revised (line 209-212; line 233-236).
5. Page 8, end of subsection 3.4: considering the objective of the paper (study of the thyristors’ breakdown risk), a conclusion should be drawn on the influence of load resistance on the risk of breakdown for thyristors.
Response: It has been revised (line 254-255).
6. Page 9, conclusion in lines 241-244: this conclusion is established exclusively from the analysis of the shape of thyristors’ current curves in the case of Type-II PPS topology. It would be correct to draw the conclusion by comparing curves similar to those in Figures 6,7 and 8, but constructed for type-II PPS topology.
Response: A new content is added, see the article for details (line 265-300).
Attached please find the revised version, which we would like to submit for your kind consideration. We would like to express our great appreciation to you and reviewers for comments on our paper. Looking forward to hearing from you.
Once again, thank you very much for your comments and suggestions.
Best regards!
Name: Jiufu Wei
E-mail: wjfnjust@163.com
Name: Zhenxiao Li
E-mail: lizhxnjust@126.com
Name: Baoming Li (corresponding author)
Round 2
Reviewer 1 Report
See attached file

Author Response
Thank you for approving our job!Reviewer 2 Report
All reviewer's comments are corrected completely.
Author Response
Thank you for approving our job!
Reviewer 3 Report
The paper was improved.
Some small suggestions below:
Line 82: Please spell ‘FRD Di’
Line 87: please, add typical value for load resistance
What are typical value of the capacitor Ci and inductances Li and Lp (pp 3)
Line 110: Why Lp could be ignored since smaller than Li? It is series with a capacitance and not with Li
Line132 Ho the total resistance and total capacitance are computed?
Line 184 and pp10-14 current ‘carves’? What is carves for authors?
Fig. 5 (b) Please add a zoom of the image around lower value of current. Here only three case are evidenced but legend reports 10 cases
Please add a survay of the possible damage in the devices and corresponding evaluation approach. What is occured in the experiment related to the all possible damages?
Author Response
Dear reviewer,
On behalf of my co-authors, we appreciate you very much for your positive and constructive comments and suggestions on our manuscript entitled “Investigation of Reverse Recovery Current of High-power Thyristor in Pulsed Power Supply” (Manuscript ID: electronics-869933). We have studied your comments carefully and have made revision which marked in red in the paper. The main responses to your comments are as following:
1) Line 82: Please spell ‘FRD Di’.
Response: We are sorry for our negligence. It has been revised (line 82).
2) Line 87: please, add typical value for load resistance. What are typical value of the capacitor Ci and inductances Li and Lp (pp 3)?
Response: Their typical values are given in the paper according to your opinion (line 88-89, 101-102).
3) Line 110: Why Lp could be ignored since smaller than Li? It is series with a capacitance and not with Li.
Response: The slope of i1(t) is dominantly determined by Vp(t) and Li in such case. The typical inductance Li is in the range of 5~40 μH. The typical value of Lp is in the range of 0.05~0.2 μH. Therefore, Lp is relatively small. Strictly, Lp is parasitic and cannot be ignored. However, in the early stage of PFU design, it is easier for designers to know Li and Vp(t), which is helpful for them to estimate the slope of i1(t) and select components of PFU or determine the design schemes.
4) Line132: How the total resistance and total capacitance are computed?
Response: We have re-written this part, see the article for details (line 120-127, 141-143).
5) Line 184 and pp10-14 current ‘carves’? What is carves for authors?
Response: Thank you for your careful review. We are sorry for our negligence in misspell words. It has been revised(line 192-194; pp 7-14).
6) Fig. 5 (b) Please add a zoom of the image around lower value of current. Here only three cases are evidenced but legend reports 10 cases.
Response: Fig. 5 (b) shows the current curves of fast recovery diodes. The lower value of the current is zero, which occurs in fast recovery diodes D3-D9. In other words, the fast recovery diodes D3-D9 are not turned on in such a case due to the pulse-shaping inductors L3-L9 are still in forward discharging phases when sustaining reverse voltages, which cause them to fail to enter freewheeling phases. In Fig. 7 (c), it is found that the voltage wave of the load side is flat-topped. Therefore, the pulse-shaping inductor Li in forward discharging phase sustains forward voltage from energy-storage capacitor Ci and reverse voltages from other discharging PFUs. The fast recovery diodes D3-D9 can be turned on by increasing the discharge time intervals. However, in EML experiment, it is not reasonable to increase the discharge time intervals all the time if considering the initial speed of armature. Therefore, parts of the fast recovery diodes are not turned on in some cases are possible.
7) Please add a survey of the possible damage in the devices and corresponding evaluation approach. What is occurred in the experiment related to the all possible damages?
Response: Thank you for your good suggestions. However, it's not realistic for us to investigate all possible damages in experiment currently. The thyristors are specially designed and the cost is high. The thyristors in a PFU is about 50000 RMB. There are currently 320 PFUs in our lab, and the cost of each PFU is about 500000 RMB. We don’t have enough money to add a survey of the possible damages in the devices in experiment currently. We will make all possible damages present in experiment if we have enough money in the future. We would be greatly appreciated if you can understand our present situation.